# Recent Developments about Conductive Polymer Based Composite Photocatalysts

**DOI:** 10.3390/polym11020206

**Published:** 2019-01-24

**Authors:** Sher Ling Lee, Chi-Jung Chang

**Affiliations:** Department of Chemical Engineering, Feng Chia University, 100, Wenhwa Road, Seatwen, Taichung 40724, Taiwan; sherlinglee0209@gmail.com

**Keywords:** conductive polymer, photocatalyst, synergic effect, photogenerated electron, photo-corrosion, antibacterial, degradation, hydrogen production

## Abstract

Conductive polymers have been widely investigated in various applications. Several conductive polymers, such as polyaniline (PANI), polypyrrole (PPy), poly(3,4-ethylenedioxythiophene) (PEDOT)), and polythiophene (PTh) have been loaded with various semiconductor nanomaterials to prepare the composite photocatalysts. However, a critical review of conductive polymer-based composite photocatalysts has not been available yet. Therefore, in this review, we summarized the applications of conductive polymers in the preparation of composite photocatalysts for photocatalytic degradation of hazardous chemicals, antibacterial, and photocatalytic hydrogen production. Various materials were systematically surveyed to illustrate their preparation methods, morphologies, and photocatalytic performances. The synergic effect between conductive polymers and semiconductor nanomaterials were observed for a lot of composite photocatalysts. The band structures of the composite photocatalysts can be analyzed to explain the mechanism of their enhanced photocatalytic activity. The incorporation of conductive polymers can result in significantly improved visible-light driven photocatalytic activity by enhancing the separation of photoexcited charge carriers, extending the light absorption range, increasing the adsorption of reactants, inhibiting photo-corrosion, and reducing the formation of large aggregates. This review provides a systematic concept about how conductive polymers can improve the performance of composite photocatalysts.

## 1. Introduction

Nowadays, organic pollutants have caused serious water pollution problems. Prior studies show that the photocatalytic degradation process using semiconductor photocatalysts is one of the efficient and green solutions to resolve these problems. Besides, semiconductor photocatalysts can be utilized for photocatalytic hydrogen production and antibacterial applications. Nevertheless, the fast recombination of photoexcited charge carriers and narrow light absorption region limit the efficiency of semiconductor photocatalyst. It was reported that incorporation of conductive materials, such as Ni foam [1,2], metal wire mesh [3,4], and graphene [5,6] can effectively solve the above-mentioned problems and improve the photocatalytic activity. The photogenerated electrons can be transferred to conductive materials, which act as electron acceptors to effectively hinder the recombination of photogenerated electron–hole pair.

In comparison with the above mentioned conductive materials, the advantage of the conductive polymers is their processability, especially through the solution process. Conductive polymers have been widely investigated in various applications, such as solar cells [7], transistor [8], thermoelectric power-generation [9], microwave absorption [10], electrochromic devices [11], photocurrent generation [12], supercapacitors [13], and light emitting diodes [14,15,16]. The composite photocatalysts can be prepared by incorporating conductive polymers (especially polyaniline, polypyrrole, PEDOT, and polythiophene) with semiconductor nanomaterials. Since conductive polymers can provide matched band structures with other inorganic semiconductors, they can reduce the recombination of photogenerated electron-hole pairs for these composite photocatalysts. In this review, compositions, preparation methods, testing conditions, possible mechanism, and improvement in photocatalytic efficiency of conductive polymers based composite photocatalysts were studied.

## 2. Polyaniline Based Photocatalysts

A large portion of the research about conductive polymer based composite photocatalysts are polyaniline based composites. PANI can combine with the semiconductor nanomaterials, such as bismuth based nanomaterials (BiPO_4_, Bi_12_O_17_C_l2_, BiOCl, BiOBr, BiOI), titanium oxide, ZnO, CdS, MnFe_2_O_4_, CoFe_2_O_4_, and Ag_3_PO_4_ to prepare the composite photocatalysts.

### 2.1. Photodegradation

The type and structure of materials used, synthesis method, light source, model pollutants, degradation performance, and mechanism for the polyaniline based composite photocatalysts were discussed as follows.

#### 2.1.1. Polyaniline-Bismuth Composites

Xu et al. reported that with the incorporation of PANI on bismuth oxyhalide can significantly improve the separation of photoexcited charge carriers and extend the light absorption region [17]. Ciprofloxacin (CIP) can be decomposed by PANI/Bi_12_O_17_C_l2_ composites as the degradation efficiency is three times higher than pure Bi_12_O_17_C_l2_ (Figure 1). In addition, this PANI/Bi_12_O_17_C_l2_ was 1.45 times more successful in removing Rhodamine B (RhB) than pristine Bi_12_O_17_C_l2_. The results indicated that the loading of PANI will not only extend light response range but also enhance the visible light-driven degradation performance of CIP. In addition, the degraded products of CIP were proved to be low-toxic to *E. coli*. (Figure 1) Proposed mechanism for photocatalytic degradation reaction by PANI/Bi_12_O_17_C_l2_ photocatalysts was shown in Figure 2. Since the CB of Bi_12_O_17_C_l2_ is less negative than lowest unoccupied molecular orbital (LUMO) potential of PANI, the photoexcited electrons of PANI are able to be transferred to the CB of Bi_12_O_17_C_l2_, and hence produce •O_2_^−^ radicals by reacting with O_2_, leading to high photocatalytic activity.

It has been proven that BiPO_4_ tends to be another promising photocatalyst which shows more attractive activity when irradiated under UV light than TiO_2_-P25, which is a good photocatalyst due to its abundance and stability. Although BiPO_4_ activity is limited under visible light irradiation, the limitations can be overcome by incorporating PANI to form heterostructure composites. The enhancement is reported by Yu et al., where the degradation efficiency under irradiation of visible light increased significantly from 23.8% to 87.3% after the co-loading of PANI layer on the surface of BiPO_4_ [18]. It is deduced that the increase in activity may be due to effective separation of photogenerated electron-hole pairs and more visible light absorption after the loading of PANI. In short, the heterostructural BiPO_4_-PANI composite photocatalyst is proved to be a promising photocatalyst to achieve high photocatalytic activity.

Besides extending the light absorption to visible light region, the loading of PANI with bismuth oxyhalide photocatalysts will also form a synergistic effect between BiOI and PANI. For instance, the research done by Yan et al. has provided evidence on the photocatalytic effectiveness of BiOI after loading with PANI [19]. The modification of BiOI with PANI brought about a more effective separation process of electron-hole pairs, which hinders the recombination of charge carriers. The RhB degradation efficiency by PANI/BiOI (12.5 wt %) is 4 times higher than the pristine BiOI. With a simple co-precipitation method, an effective photocatalyst composed of PANI/BiOI with 91% degradation efficiency of RhB can be fabricated. Similar synergistic effect can also be observed in other studies, such as loading of PANI with BiOBr [20] and incorporation of PANI with BiOCl [21]. A proposed photocatalytic mechanism for PANI/BiOCl composite photocatalyst is presented in Figure 3. The absorption of visible light can induce the π–π* transition of PANI, and the excited-state electrons of highest occupied molecular orbital (HOMO) orbital is transferred to the LUMO orbital. Because of the matched band structure between π*-orbital of PANI and the CB of BiOCl, a synergistic effect was observed. The excited electrons were injected to the CB of BiOCl and then migrated to the photocatalyst surface to react with water and oxygen. Then, active superoxide and hydroxyl radicals were produced, which started an oxidization reaction with the methyl orange dye.

The loading of bismuth vanadate with PANI (molecular weight ~ 10^5^) is studied by Shang et al. where the photocatalyst is synthesize through a facile sonochemical method [22]. Pure BiVO_4_ is known to have low photocatalytic activity, as it shows low adsorption capability and inefficient transfer of electron-holes pairs. However, the loading of conducting polymer can enhance the charge transfer efficiency, leading to a rapid photodegradation rate of organic pollutants. The novel spindle-like PANI/BiVO_4_ photocatalyst is visible light sensitive, as it shows 100% photodegradation for both RhB and phenol with H_2_O_2_ added after 60 min light irradiation. The synergic effect was observed between PANI and BiVO_4_, where the photoinduced holes in the valence band of BiVO_4_ was rapidly transported to the π-orbital of PANI, while the conduction band of BiVO_4_ will attract photoinduced electrons, thus, preventing the recombination of photoexcited charge carriers. 

Table 1 exhibits the compositions, preparation methods, testing conditions, and photodegradation efficiency of various PANI/bismuth-based compounds composite photocatalyst.

#### 2.1.2. Polyaniline-Titanium Oxide

Besides facing the problem of water being seriously polluted by antibiotic agent, pharmaceutical and pesticides are also detected in the environment. According to the recent research done by Merkulov et al., the use of photocatalyst composed of polyaniline (PANI) and titanium oxide can photodegrade various pharmaceuticals and pesticides under the irradiation of UV light [23]. The photodegradation reactions are carried out under different conditions: double distilled water, river, and lake water. Among all model pollutants tested, the photodegradation activity for sulcotrione is the best of all, with degradation efficiency of 93.8% under the presence of TiO_2_/PANI-150. 

There have been numerous studies reported on the degradation of pollutants, such as organic pollutants (Bisphenol A (BPA), RhB) and harmful dyes by PANI/TiO_2_ composite photocatalyst. The photodegradation of organic pollutants is exhibited by PANI/TiO_2_-graphene hydrogel composite photocatalyst (PANI/TiO_2_-rGH), which were prepared by a process shown in Figure 4a [24]. Figure 4b presents the transmission electron microscopy (TEM) image of PANI/TiO_2_-rGH. The highest degradation efficiency of BPA is exhibited by 5% PANI/TiO_2_. The incorporation of PANI promotes the separation of electrons and holes. In addition, the enhancement is attributed to the π–π stacking interaction and formation of hydrogen bond between BPA and graphene. On the other hand, photocatalytic degradation of RhB is demonstrated in the study done by Xu et al. on composite photocatalyst PANI/grey-TiO_2_ [25]. The results showed that after 180 min of visible light irradiation, PANI/TiO_2_ can degrade RhB completely, as the loading of PANI accelerate the charge transfer between the interface of PANI and grey-TiO_2_. This result is achieved by TiO_2_ with the optimum amount of 5.0 wt % of PANI loaded, as further increase of amount of PANI will lead to decrease of hydrogen production rate. 

Gilja et al. discussed the photocatalytic performance and stability of PANI/TiO_2_ composite photocatalyst fabricated through in-situ chemical oxidation [26]. Due to TiO_2_ particle collisions, large aggregates may form and greatly decrease the photocatalytic activity of PANI/TiO_2_. However, a high concentration of aniline can help reduce the formation of large aggregates, as aniline molecule tend to create a barrier to the aggregation process of TiO_2_ nanoparticles. Under UVA irradiation, 15 wt % PANI/TiO_2_ photocatalyst showed the highest effectiveness to complete degrade Reactive Red 45 azo dye (RR45), as the degradation efficiency reached 94% and 80% for total organic carbon (TOC) removal. This effectiveness is ascribed to the synergetic effect between PANI and TiO_2_ and to the fact that PANI protected TiO_2_ surface from intermediates blockage. The synergistic effect is also reported in the study done by Razak et al., where a reusable immobilized TiO_2_-PANI composite photocatalyst plate is fabricated [27]. As TiO_2_/PANI/ENR/PVC is immobilized on the plate, it increases the specific surface area thus improves the degradation efficiency of RR45 dye, where photocatalytic activity reaches as high as 85% after 60 min of visible light irradiation. 

Recently, Deng et al. performed the study on the incorporation of PANI with mesoporous single crystal TiO_2_ (MS–TiO_2_) microsphere [28]. The research proved that with the loading of PANI, the photocatalytic activity of TiO_2_ increased significantly to 99.8% in 120 min and 99.5% in 150 min for RhB and methylene blue (MB), respectively. The enhancement is three times higher than pure MS–TiO_2_ as the unique mesoporous structure exhibits higher specific surface area, which allows more PANI to be loaded on the surface of MS–TiO_2_, as well as promotes multiple reflection in the photocatalyst. Similar trend of synergistic effect also exhibited in other PANI/TiO_2_ composite photocatalysts, such as sulfonated-PANI@defective-TiO_2_ (s-PANI@m-TiO_2_) [29], PANI nanotube @ TiO_2_ [30], three dimensional flowerlike TiO_2_/PANI [31], PANI-sensitized TiO_2_ [32], and TiO_2_/PANI fabricated by in-situ polymerization [33].

Figure 5a shows the antibacterial performances of different photocatalysts against *E. coli* [30]. The result reveals that the PANI–TiO_2_ composite photocatalyst shows effective inhibition of *E. coli* in comparison with pristine PANI and TiO_2_. Figure 5c shows a TEM image of *E. coli* treated with the PANI–TiO_2_ composite photocatalyst. The cell structure of the bacteria treated with PANI–TiO_2_ photocatalyst was damaged drastically as compared to image of *E. coli*. Figure 5b. Essential metabolites were released when the membrane was disrupted. Table 2 presents the compositions, preparation methods, testing conditions, and photodegradation efficiency of PANI/TiO_2_ composite photocatalysts. 

#### 2.1.3. Polyaniline with Other Compounds

Prior literatures showed that zirconia nanostructures possess good photocatalytic capabilities for the degradation dyes [34,35], phenol, and its derivatives [36,37]. In order to further improve the efficiency of zirconia photocatalyst, PANI is incorporated, as PANI can effectively absorb coefficients in the visible light spectrum, as well as accelerate the mobility of charge carriers. Carević et al. fabricated a novel ZrO_2_/PANI nanocomposite via oxidative polymerization of aniline with ammonium peroxydisulfate (APS) in water with the presence of ZrO_2_ nanoparticles [38]. The nanocomposite photocatalyst containing the largest amount of PANI exhibited the highest photocatalytic activity, where the degradation activity of trichlorophenol (TCP) is better than commercial Degussa TiO_2_ within the 100 min of illumination, carbon nitride, and zirconia.

Carbon nitride photocatalyst has been gaining attention of researchers due to its 2D structure, tunable electronic structure, and good chemical stability [39]. However, the application of carbon nitride in photocatalytic activity is limited, as it displays rapid recombination of photogenerated charge carrier and low surface area. Jiang et al. proposed the loading of polyaniline with carbon nitride nanosheets to fabricate novel 3D hierarchical structured photocatalyst [40]. The research showed that the photodegradation efficiency of carbon nitride nanosheets (CNNS) reached 92.5%, which was slightly higher than that of PANI/CNNS-5% (about 89.1%). The lower photodegradation efficiency of PANI/CNNS-5% is ascribed to the CN layers being bonded together after the incorporation of PANI into the layers of CNNS, thus having smaller Brunauer, Emmett, and Teller (BET) specific area compared to pristine CNNS. However, PANI/CNNS-5% is proved to be more advantageous as it displayed higher phenol removal rate than CNNS, due to its excellent separation of interfacial charge and photoelectrochemical performance. Figure 6 shows the schematic illustration of the synergistic removal of organic pollution by PANI/CNNS composite photocatalysts. Pollutants can be adsorbed and concentrated into the photocatalyst with 3D hierarchical nanostructure, which can hinder the stacking of subunits. Based on the band structure, the photoexcited electrons on the LUMO of PANI can transfer to the LUMO of CNNS. The photoexcited holes on the HOMO of CNNS can migrate to the HOMO of PANI and travel through the 3D structure.

In addition to good photocatalytic capabilities, it is also vital to fabricate the photocatalyst, which is recyclable. In order to synthesize magnetically recyclable photocatalysts, Zeng et al. incorporated MnFe_2_O_4_ with PANI via facile in-situ oxidative polymerization method [41]. The combination of PANI and MnFe_2_O_4_ improves the photocatalytic activity by providing more surface area and smaller particle size, and thus effectively hinders the recombination of photoinduced charge carriers. A similar magnetically recyclable photocatalyst is synthesized by Kim et al. with cobalt ferrite and PANI nanofibers via the electrospinning technique and an in-situ chemical oxidative polymerization [42]. The hollow CoFe_2_O_4_–PANI nanofibers coated with PANI nanograins forms a heterojunction between the interface of CoFe_2_O_4_ and PANI, and thus increases the absorption of visible light, as well as accelerates charge separation. The result evidenced the photocatalytic degradation of methyl orange by CoFe_2_O_4_–PANI hollow nanofibers is 80 folds higher than that pristine CoFe_2_O_4_. In comparison with the pure photocatalyst, the loading of PANI with magnetic photocatalyst will greatly enhance the photodegradation activity. It is also worth noting that both MnFe_2_O_4_/PANI and CoFe_2_O_4_/PANI do not exhibit any significant loss of photocatalytic activity, even after three recycles.

Recently, fabrication of ternary nanocomposite has gained much attention, as it enhances the synergetic effect among all three components. For example, Wu et al. synthesized a ternary photocatalyst of zinc oxide/reduced graphene oxide (rGO)/PANI and examined the photocatalytic activity using methyl orange (MO) as the model pollutant [43]. It is found that the enhancement in photodegradation efficiency is 6 times higher than pristine ZnO within 60 min of UV irradiation. It is deduced that the enhancement is ascribed to rGO, which promotes the transfer of photoinduced electrons, as well as PANI, which increases both the adsorption of dyes and absorption of UV light. In short, the presence of synergistic effect between ZnO/rGO/PANI interfaces can greatly improve the photodegradation capabilities of PANI composites. 

The enhancement of photocatalytic activity with the loading of PANI is also exhibited in the research done by Bu and Chen, where PANI is fabricated with silver and silver phosphate [44]. Although Ag_3_PO_4_ exhibits good photocatalytic properties under visible light exposure, the photo-corrosive behavior in the presence of light limits its practical application in the environment. Nevertheless, the loading of PANI significantly improves the photostability of Ag_3_PO_4_, as the presence of π-conjugated long chain in PANI traps the photoinduced holes, and thus hinders self-oxidation of Ag_3_PO_4_. The enhancement is evident in the result where the photodegradation of RhB reaches more than 95% within 5 min light irradiation, which is about 4 times better than that of pristine Ag_3_PO_4_. 

The composite photocatalyst PANI-graphitic carbon nitride (*g*–C_3_N_4_) with different structures is also investigated [45]. PANI nanorods (molecular weight 5 × 10^4^) are grown on the surface of *g*–C_3_N_4_ sheets via dilute polymerization method under very low temperature of −20 °C. The hierarchical CN–PANI composite photocatalyst exhibited better photodegradation effectiveness on methyl orange (MO), about 99.8% after 30 min visible light illumination. The enhancement undergone by CN–PANI is four times higher than that of pristine *g*–C_3_N_4_, because the loading of PANI on the CN surface hinders the recombination of photoinduced charge carriers. In short, the photocatalytic capability can be improved significantly by incorporating PANI with carbon nitride. 

The combination of PANI with zinc oxide can enhance the photocatalytic capabilities because of the significant synergic effect between defective ZnO and PANI. Based on the research done by Pei et al., the activity of PANI/ZnO photocatalyst is 2.5 times higher than pristine ZnO for the photodegradation of MO under ultraviolet illumination [46]. In addition, for the removal of pollutants that are more toxic and stable, like 4-chlorophenol (4-CP), PANI/ZnO nanocomposite still showed great photocatalytic activity, as it reported nearly 82% of total organic carbon species removed after 120 min UV irradiation. Besides the combination with defective ZnO, Eskizeybek et al. proposed the fabrication of PANI/ZnO with the use of PANI homopolymer and ZnO nanoparticles [47]. When the PANI/ZnO composite photocatalyst is examined under natural sunlight irradiation and UV light, it is shown that the photocatalyst is more effective under natural sunlight, where the degradation efficiency of MB and malachite green (MG) after sunlight irradiation are 97% and 99%, respectively. In short, the high photocatalytic activity results from the synergic effect between the ZnO and PANI interface, which leads to improved transfer of photoinduced charge. 

Based on the study done by Zhang and Zhu on cadmium-chalcogen-based PANI photocatalysts, the hybrid effect between CdS and PANI plays a vital role in enhancing photocatalytic activity, as well as preventing the photo-corrosion of CdS [48]. The results show a 1.5 times higher photodegradation rate of MB for PANI monolayer-hybrid cadmium sulfide photocatalyst than that of pristine CdS. Although the incorporation of PANI on CdS can accelerate the rate of photodegradation, higher PANI content than the optimum amount of 5% will lead to a decrease, as the accumulated PANI will hinder the adsorption of MB. Table 3 summarizes the compositions, preparation methods, testing conditions, and photodegradation efficiency of other polyaniline based composite photocatalysts.

### 2.2. Photocatalytic Hydrogen Production 

#### Polyaniline with Other Compounds

Among wide range of semiconductors, cadmium sulfide is widely used in photocatalytic hydrogen production, as it has a small band gap of 2.4 eV, which enables it to become responsive towards visible light. However, due to the lower surface bond energy of CdS nanostructures, CdS is less stable during the photocatalytic hydrogen production reaction. The effects of different conductive polymers (PANI, PPy, and poly(3,4-ethylenedioxythiophene) (PEDOT)), on the mechanism of the conductive polymers@CdS core-shell nanorods were studied. The HOMO and the LUMO energy levels of these three conducting polymers derived from the cyclic voltammetry (CV) curves were shown in Figure 7. In comparison with the PEDOT@CdS photocatalyst, Wang et al. [49]. reported that PPY@CdS and PANI@CdS photocatalysts possess larger driving force for the injection of photoexcited holes into the HOMO of the conducting polymer shell [50]. The modification of CdS nanorods by PPY and PANI results in not only the separation of photoexcited electron-hole pairs, but also photo-corrosion inhibition. PANI exhibited the most significant enhancement of hydrogen production rate as high as ~9.7 mmol h^−1^ g^−1^. Table 3 displays the activity of PANI/ZnO composite photocatalyst under the irradiation of visible light.

Xu et al. studied the composite photocatalysts consisting of grey-TiO_2_ nanomaterials and conductive PANI, which were prepared by the reduction of white-TiO_2_. The composite was synthesized through the in-situ polymerization of aniline in the presence of grey-TiO_2_ nanomaterials. The highest hydrogen production rate of 1.79 mmol g^−1^ h^−1^ was almost 2.5 times higher than the pristine grey-TiO_2_ [25]. In addition, the composite photocatalyst also exhibits enhanced photocatalytic degradation activity, about 100% of Rhodamine B can be degraded within 3 h. The enhanced activity is ascribed to the effective charge transfer between the interface of cocatalyst grey-TiO_2_ and PANI, which in turn hampers the recombination of electrons-holes pairs. 

Moreover, based on prior study done by Kato and Kudo in 1998, it was revealed that alkali tantalate is one of the most promising photocatalysts for the photo-decomposition of water under UV light irradiation [51]. Zielińska et al. reported the loading of PANI on NaTaO_3_ via oxidative polymerization in acidic solution containing sodium tantalate [52]. After 120 min of UV light illumination, the hydrogen production of PANI/NaTaO_3_ reached about 2 times higher than that of pure NaTaO_3_. The enhancement is brought about by the efficient charge separation process, which can be showed through photoluminenscene (PL) spectrum. In comparison with pure NaTaO_3_, PANI/NaTaO_3_ exhibits lower PL spectrum, which suggests a slower recombination of photoexcited electron-hole pairs in PANI/NaTaO_3_ photocatalyst, thus leading to a better activity. 

In addition, the effects of incorporating zinc sulfide with PANI via solvothermal method was studied [53]. The hydrogen production of ZnS/PANI with 40 wt % of PANI reached as high as 6750 μmol h^−1^ g^−1^, which is about 4 times higher than the pristine ZnS. It results from the superior properties of PANI, where effective transfer of photoinduced holes and efficient separation of electron-hole pairs can occur in the composite photocatalyst. In addition, the photocatalytic activity is enhanced, as PANI-ZnS photocatalysts possess a good dispersion stability in the aqueous sacrificial solution, where a better contact between photocatalyst and sacrificial agent is present. 

In recent years, ZnO based photocatalyst has attracted many researchers, as it has significantly higher quantum efficiency than TiO_2_, together with good photochemical properties. However, due to its fast recombination of electron-holes pairs and inability to be use under visible sunlight, ZnO is modified with the addition of additives. Nsib et al. [54]. showed the incorporation of polyaniline and nickel doping can improve ZnO morphology. The p-n heterojunction forms between ZnO and NiO in the Ni-doped ZnO nanomaterials. The highest amount of hydrogen produced by Ni_0.1_ZnO_0.9_/PANI_10_ reached 558 μmole h^−1^, higher than that for pure ZnO (178 μmole h^−1^). The loading of PANI can effectively enhance the hydrogen production rate by widening the photo-response region towards the visible light region and narrowing the band gap of ZnO. Besides, the type of the sacrificial agent can affect the photocatalytic H_2_ generation activity and changes in the following order: S_2_O_3_^2−^ > S^2−^ > C_3_H_7_OH.

Recently, incorporation of multiple active nanomaterials into hierarchical structures is one of the important research issues. It is reported that the activity can be enhanced significantly with the loading of both polymeric (or semiconductor) photocatalyst and transition metals with PANI. For example, a recent research study about the incorporation of carbon nitride and silver with PANI resulted in a *g*-C3N4–Ag/PANI photocatalyst with hydrogen production rate of 210.73 μmol h^−1^ mg^−1^ [55]. The enhancement is attributed to the sufficient amount of visible light absorbed and large amount of active sites provided. Moreover, the surface plasmon resonance (SPR) and band structure difference displayed by Ag nanoparticles in the ternary heterostructures improves the spatial separation and transfer of photogenerated charge carriers at the heterojunction interface. The improved activities result from good adsorption property, high surface areas, and excellent separation of photoexcited charge carriers across the interface.

Besides the ternary hetero-structured PANI photocatalysts with metal-free polymeric photocatalyst and transition metal, Sasikala et al. reported the combination of PANI with MoS_2_ and CdS semiconductor for photocatalytic hydrogen production [56]. The presence of both MoS_2_ and PANI enhanced the visible light absorption ability of MoS_2_–PANI–CdS photocatalysts and improved the lifetime of photogenerated electron-hole pairs. The hydrogen generation activity increases with increasing loading amount of MoS_2_ and PANI. Then, an optimum amount of 4% MoS_2_ and 5% PANI is reached. Further increase of MoS_2_/PANI will decrease the photocatalytic activity. The increased visible light absorption and improved lifetime of the photoexcited electron-hole pairs leads to improved activity of the photocatalysts. Table 4 shows the details of photocatalytic hydrogen production of polymeric PANI-based composite photocatalysts.

## 3. Polypyrrole Based Photocatalysts

Some polypyrrole-(PPy) based composite photocatalysts were reported. PPy can combine with the semiconductor nanomaterials, such as BiOBr, BiOI, titanium oxide, ZnO, CdS, AgPMo_12_, ZnIn_2_S_4_, Fe_2_O_3_, and Bi_2_WO_6_, to prepare the composite photocatalysts. The pristine polypyrrole and the composite photocatalysts can be used for photodegradation of organic pollutant and photocatalytic H_2_ production.

### 3.1. Photodegradation

#### 3.1.1. Pristine Polypyrrole

Polypyrrole is another important conductive polymer for the fabrication of polymeric photocatalysts. Yuan et al. reported the study of photocatalytic activity of pristine polypyrrole nanostructures (PPy–NS) on photodegradation of organic pollutants, such as phenol and MO [57]. The study showed that the photocatalytic performance of PPy was affected by the fabrication method and type of light irradiated. For example, PPy–NS-c synthesized through chemical oxidation showed better photodegradation capabilities for phenol (~100%) than that of PPy–NS-γ and PPy-bulk after irradiation of UV light for 4.5 h. However, under 5 h of visible light illumination, PPy–NS-γ exhibited 2 times higher photodegradation rate of phenol (20%) than PPy–NS-c, and 4 times higher than that of PPy-bulk. This study presented the vital factor that determines the application of PPy is the nano-structuration of CPs, as the structure changes the surface area of the CP and the band gap energy. 

#### 3.1.2. Polypyrrole Based Composites

Cadmium sulfide is one of the most promising photocatalysts that is active in the visible light wavelength, as it has a low band gap energy of 2.3 eV. However, the application is limited due to the occurrence of self-photo-corrosion and poor stability of CdS. With the loading of conductive polymers, such as PPy, the π–π conjugated orbital of PPy will slow down the rate of carrier recombination and widen the visible light absorption region. For example, the study done by Shan et al. on the incorporation of PPy with CdS to form nanocomposite photocatalysts brought about 2 times faster photodegradation rate than that of pure CdS nanoparticles [58]. The presence of PPy layer can stabilize CdS by acting as the protective layer that hinders the ionization, thus reducing the corrosion of CdS.

Titanium dioxide nanoparticles are well known to be an excellent photocatalyst due to their excellent chemical and photostability, less toxicity, and low cost. A large band gap energy of 3.2 eV limits the application of titanium dioxide under visible light condition. Nevertheless, Baig et al. reported the coupling of PPy decreased the band gap of TiO_2_ and ensured the visible light-driven photocatalytic activity [59]. The study showed that the adsorption of MO is increased in PPy–TiO_2_ nanocomposite photocatalyst when compared to pure TiO_2_. The removal efficiency of 100% MO by PPy–TiO_2_ can be achieved after 60 min visible light irradiation, whereas TiO_2_ only managed to reach 55% of MO removal efficiency. Hence, it is undeniable that with the formation of heterojunction between PPy and TiO_2_, it will accelerate the transfer of electron-hole pairs and hinders the recombination of photogenerated charge carriers, thus, forming a stable and effective photocatalyst. A similar trend of photodegradation is also evidenced in the study done by Gao et al., where Rhodamine B is used as the organic pollutant instead to examine the photocatalytic activity of PPy/TiO_2_ nanocomposites [60].

Liu et al. reported the loading of PPy and Ag nanoparticles with BiOBr in order to modify the characteristics of BiOBr [61]. The fabricated BiOBr–Ag–PPy nanocomposite showed 6.4 times higher photocatalytic degradation than that of pristine BiOBr, owing to the heterojunction structure between the composite photocatalyst. Through electrochemical impedance spectroscopy (EIS) Nyquist plots of pure BiOBr, BiOBr–Ag, BiOBr–PPy, and BiOBr–Ag–PPy (Figure 8), it suggests that with the coupling of silver nanoparticles and PPy, a lower resistance of charge transfer is achieved, thus leading to an efficient separation of electron-holes pair. However, an optimum amount of PPy should be added in order to obtain the maximum photodegradation rate, as extra PPy will accumulate on the surface of BiOBr, thus reducing the adsorption of organic pollutants. Hence, the degradation conversion of BiOBr is enhanced with the presence of both metal nanoparticles and PPy, because the type II heterojunction structure will hinder the carrier recombination and accelerate the separation of electron-hole pairs. 

In addition to high degradation conversion of organic pollutants, it is also very vital to fabricate photocatalysts that are environmentally friendly, including having high recyclability. In order to design a photocatalyst that is able to be easily separated from the suspension system and be reused numerous times, Yang et al. demonstrated the incorporation of magnetic material with a semiconductor photocatalyst [62]. Modified iron (III) oxide (M–Fe_2_O_3_) and PPy is loaded with TiO_2_ to synthesize a photocatalyst that can be easily induced in the visible light region. The degradation efficiency of methyl orange under UV radiation for 90 min reached as high as 90.4%. However, the photodegradation efficiency is much lower in the visible light region (59.3%). This difference is ascribed to the insufficient energy for the separation of electron-hole pairs under solar radiation, as only 5% of UV light is accounted in solar radiation.

Xu et al. tried to improve the capability for the photocatalytic degradation of organic pollutants by loading the conducting polymer with bismuth oxyhalides [63]. PPy has been incorporated with photocatalysts, as their unique π-conjugated electron system enables efficient charge transfer between PPy and bismuth oxyhalides. The BiOI nanosheets loaded with PPy showed significant enhanced activity of about six times higher than that of pristine BiOI in the photocatalytic degradation of RhB after being irradiated for 300 min. It is proposed that the synergetic effect formed between the intimate contact of BiOI and PPy ease the mobility and separation of photoexcited charge carriers, thus resulting in higher photocatalytic activity.

Moreover, PPy can absorb the visible light, thus the addition of PPy is considered to be one of the most promising approaches for increasing the degradation efficiency of toxic pollutants. For instance, Zhang et al. suggested the coupling of PPy with Bi_2_WO_6_ will be able to synthesize a visible light induced composite photocatalyst [64]. Comparing the high resolution Raman spectra of Bi_2_WO_6_/PPy and Py in the region of 100–180 cm^−1^, a red shift in wavenumber is observed. The red shift suggests that there is a covalent-like interaction between PPy and Bi_2_WO_6_. It helps to accelerate the separation of electron-hole pairs. In short, the presence of PPy in Bi_2_WO_6_ enables the composite photocatalyst to achieve photodegradation efficiency of about 100% within 120 min of light irradiation, whereas pure Bi_2_WO_6_ only reaches 72.4% within the same irradiation time.

Gao et al. reported the study of ternary zinc sulfide as a visible light induced photocatalyst, where the photodegradation capability is significantly enhanced with the incorporation of conducting polymer, such as PPy [65]. The limitation encountered by ternary zinc sulfide is resolved by the loading PPy to serve as an efficient electron provider, which in turn accelerates the separation process and suppresses the recombination of charge carriers. The results indicate that a complete degradation of organic pollutant chloramphenicol (CHL) by PPy–ZnIn_2_S_4_ composite photocatalyst with optimum amount of PPy is 2 times faster than pure ZnIn_2_S_4_ photocatalyst. Moreover, PPy–ZnIn_2_S_4_ also showed better performance in total organic compound (TOC) removal of 48.5% after 180 min degradation than that of ZnIn_2_S_4_ (23.5%). Based on the analysis of intermediates formed during photodegradation, it is deduced that there are lesser and simpler intermediates present in PPy–ZnIn_2_S_4_ system, hence increasing the photodegradation rate and TOC removal rate.

In addition to semiconductor photocatalysts, polyoxometalates (POMs) can act as the template of metallacycle compound photocatalyst [66]. As POM contains oxygen-rich surfaces to ease the synthesis of inorganic-organic hybrid composites (POM@CPs), however the insensitivity of POM@CPs in visible light spectrum limits its photocatalytic activity. With the loading of polypyrrole on the surface of POMs-templated metallacycle hybrid compound [Ag_6_trz_6_][H_3_PMo_12_O_40_]_2_·6H_2_O (AgPMo_12_), the photocatalytic activity towards RhB is enhanced by 11 folds of pure AgPMo_12_ and about 8 folds of pristine PPy. In addition to the improvement of photocatalytic performance, the modified PPy/AgPMo_12_ tends to be more stable. It remains active, as no great decrease is observed after 5 rounds of recycle photocatalytic tests. Hence, in comparison with both pure PPy and AgPMo_12_, the introduction of PPy in AgPMo_12_ truly creates a significant enhancement in the photodegradation rate of organic pollutants. Table 5 presents the compositions, preparation methods, testing conditions, and photodegradation efficiency of polypyrrole-based composite photocatalysts. 

### 3.2. Photocatalytic Hydrogen Production

In a recent research study conducted by Li et al., it is demonstrated that the coupling of both noble metal and conducting polymer will enable semiconductor photocatalysts to achieve high photocatalytic activity [67]. Conducting polymer is a promising candidate to modify the light absorption spectrum of photocatalyst, owing to its π-conjugated electron system. On the other hand, the coupling of noble metal enables photocatalysts to be more efficient, as the noble metal nanoparticles accumulated on the surface of the photocatalyst will serve as electron trappers, and thus inhibit the recombination of photogenerated charge carriers. The hydrogen production rate of TiO_2_–Pd–PPy increased about 3.3 times more than that of pristine TiO_2_–0.5Pd, attributed to the synergetic effect between the ternary composite photocatalyst. This enhancement is evident in the photoluminescence (PL) emission spectra of TiO_2_, TiO_2_–0.5Pd, and TiO_2_–0.5Pd–0.6PPy, where TiO_2_–0.5Pd–0.6PPy exhibits lower PL intensity, thus reflecting better inhibition for the recombination of electron-hole pairs. At the same time, the concentration of PPy must not be more than 0.6%, as beyond this optimum concentration, the photocatalytic activity decreases gradually. In short, modification of TiO_2_ by loading PPy and Pd nanoparticles indeed enhanced the photocatalytic hydrogen production activity. Figure 9 presents the energy level diagram of TiO_2_–Pd–PPy photocatalysts and the photoexcited carrier transfer processes. When irradiated by UV–visible light, PPy can absorb photons to start π–π* electron transition; the photoexcited electrons can inject into the CB of TiO_2_ and are then caught by the Pd metal. Meanwhile, when TiO_2_ is excited, the photo-excited holes at VB of TiO_2_ can migrate to the HOMO of PPy. PPy can act as a hole transporting material for the migration of photogenerated holes. When irradiated by visible light, only PPy can be excited as a photosensitizer, as PPy will inject photogenerated electrons into the CB of TiO_2_ and soon be caught by Pd. The photogenerated holes of PPy can participate in the reaction.

## 4. Polythiophene Based Photocatalyst

Only limited amounts of polythiophene (PTh) based composite photocatalysts were reported for the photodegradation of organic pollutants and the photoreduction of CO_2_.

### 4.1. Photodegradation or Photoreduction

#### Polythiophene with Other Compounds

Polythiophene (PTh) also gained much attention from researchers, as it is well known as one of the most promising conducting polymers, which significantly enhances photocatalytic activity of the combined photocatalyst. The enhancement is due to the fact that presence of PTh will increase the ability of adsorption of the catalyst and widen the light adsorption region of the catalyst towards the visible light region. The recent study by Chandra et al. showed the effectiveness of PTh when incorporated with a copper doped TiO_2_ photocatalyst [68]. The report showed that the degradation of Rhodamine B (RhB) by Cu–TiO_2_/PTh nanorods was as high as 99.4% after 75 min of visible light irradiation, while only 70.5% of RhB was degraded by pure Cu–TiO_2_ nanorods. In addition, the degradation of Orange G (OG) using Cu-TiO_2_/PTh nanorods is 1.4 times higher than pristine Cu–TiO_2_ nanorods. These enhancements of composite photocatalysts by incorporating polythiophene are ascribed to the larger surface area of photocatalysts, which allow more adsorption capacity.

Moreover, the performance of polythiophene incorporated in catalyst is also evaluated in the study done recently by Yu et al. with graphene oxide [69]. It is observed that complete photodegradation of methylene blue (MB) model pollutants is reached after 30 min of visible light irradiation, as the catalytic activity of graphene oxide/polythiophene (GO/PTh) is enhanced by 382 and 41 times than that of pristine PTh and GO, respectively. The photocatalytic capability is enhanced due to the synergistic effect between GO and PTh, good electron-transfer ability of GO, and effective light adsorption capability of PTh, which are evidenced in the analysis of UV–Vis spectroscopy and PL spectra. However, an optimum GO/Th weight ratio (1:4) is needed to obtain an efficient photodegradation rate. When GO is completely covered by PTh, it will result in less adsorption of MB. Therefore, the activity of photocatalysts can be enhanced with the loading of the optimum amount of PTh.

Furthermore, in comparison with other conducting polymers, Dai et al. reported that the performance of the photoreduction of CO_2_ can be significantly improved by loading PTh on the photocatalyst [70]. When Bi_2_WO_6_ hierarchical hollow microspheres were incorporated with PANI, PPy, and PTh, respectively, the photocatalytic CO_2_ reduction of PTh/Bi_2_WO_6_ showed the highest yield of 56.5 μmol g_cat_^−1^ methanol and 20.5 μmol g_cat_^−1^ ethanol. The total product (methanol and ethanol) yield of PTh/Bi_2_WO_6_ is 1.7 and 1.3 times higher than that of PPy/Bi_2_WO_6_ and PANI/Bi_2_WO_6_, respectively. PTh exhibits a narrow band gap and good charge mobility when modified with Bi_2_WO_6_. Hence, the order of photocatalytic activity of the as-prepared photocatalysts is Bi_2_WO_6_ < PPy/Bi_2_WO_6_ < PANI/Bi_2_WO_6_ < PTh/Bi_2_WO_6_. 

## 5. Conclusions

This review demonstrated that conductive polymers are useful to improve the performance of composite photocatalysts for photocatalytic degradation of hazardous chemicals, antibacterial, and photocatalytic hydrogen production applications, focusing on the roles of conductive polymers. The incorporation of conductive polymers can significantly improve the visible-light driven photocatalytic activity by enhancing the separation of photogenerated charge carriers and extending the light absorption range. Besides, the loading of conductive polymers with semiconductor photocatalysts also leads to a synergistic effect between conductive polymers and semiconductor nanomaterials, leading to high photocatalytic degradation efficiency. A high concentration of conductive polymers can help to reduce the formation of large aggregates, as polymer molecules tend to create a barrier to the aggregation process of photocatalysts. When the bacteria were treated with a conductive polymer-based composite photocatalyst, their cell walls and membranes were damaged drastically, and interior substances were damaged rapidly. The loading of conductive polymers can improve the photostability (inhibit the photocorrosion), as the π-conjugated long chain in the conductive polymer traps the photoinduced holes, and thus hinders self-oxidation of photocatalysts. When the conductive polymer is combined with a semiconductor photocatalyst, it plays different roles under different light irradiation. When the composite photocatalysts are irradiated by UV–visible light, the conductive polymer can act as a cocatalyst, which is excited by irradiated light and a hole transporting material. When irradiated by visible light, only the conductive polymer can be excited as a photosensitizer. This review provides a systematic concept about how conductive polymers contribute to improving the performance of composite photocatalysts. 

## Figures and Tables

**Figure 1 polymers-11-00206-f001:**
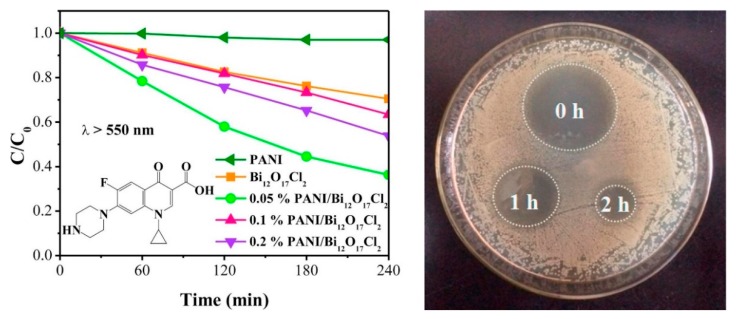
(**Left**) Visible light-driven photocatalytic degradation of CIP by various photocatalysts. (**Right**) The inhibition zone of the degraded CIP solution [17].

**Figure 2 polymers-11-00206-f002:**
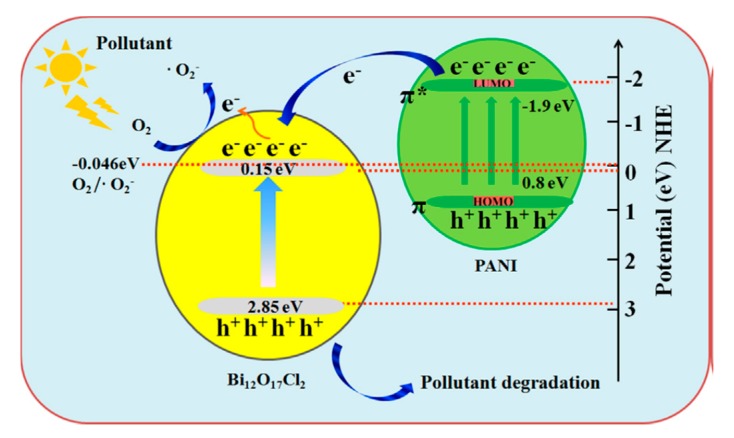
Proposed mechanism for photocatalytic degradation reaction under visible light irradiation by PANI/Bi_12_O_17_Cl_2_ composite photocatalysts [17].

**Figure 3 polymers-11-00206-f003:**
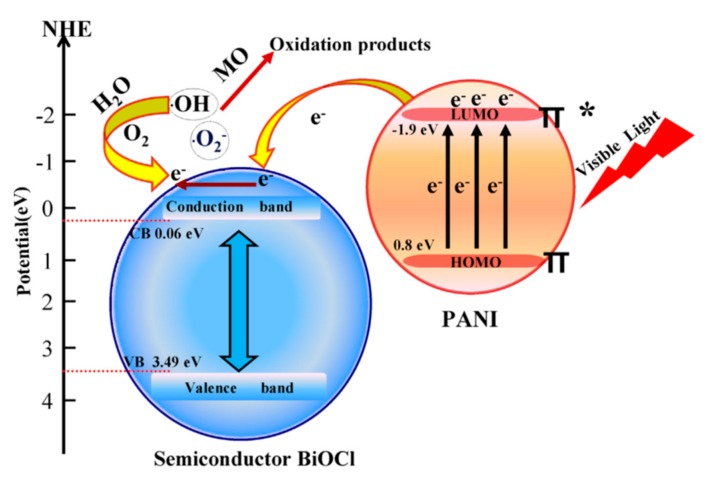
Schematic illustration of the proposed photocatalytic mechanism of PANI/BiOCl under visible light irradiation [21].

**Figure 4 polymers-11-00206-f004:**
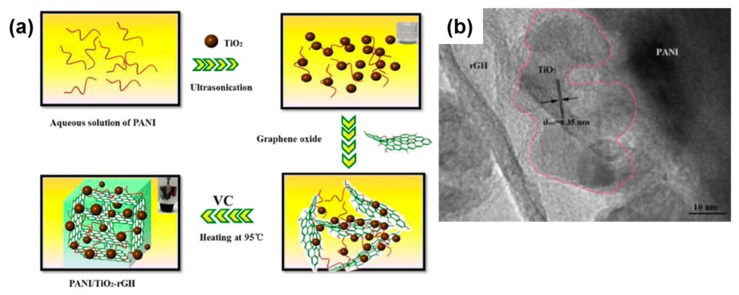
(**a**) Graphical illustration of fabrication of PANI/TiO_2_-rGH, (**b**) TEM of PANI/TiO_2_ –rGH [24].

**Figure 5 polymers-11-00206-f005:**
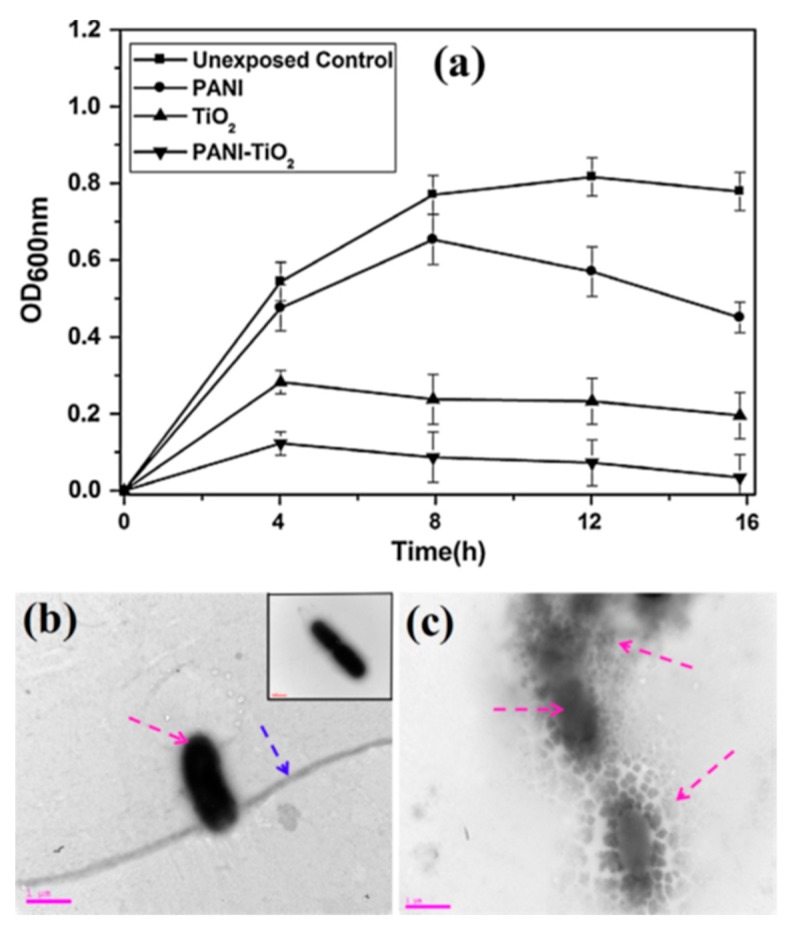
(**a**) The antibacterial performances of PANI–TiO_2_, TiO_2_, and PANI photocatalysts against *E. coli*, and TEM image of (**b**) *E. coli* (inset native *E. coli*), and (**c**) *E. coli* treated with PANI–TiO_2_ composite photocatalyst [30].

**Figure 6 polymers-11-00206-f006:**
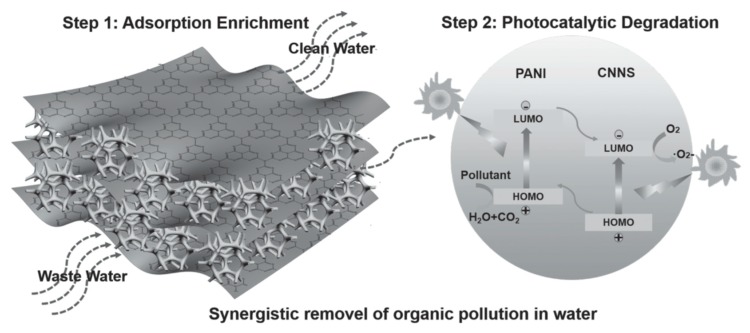
Schematic illustration of the synergistic removal of organic pollution by PANI/CNNS composite photocatalysts [40].

**Figure 7 polymers-11-00206-f007:**
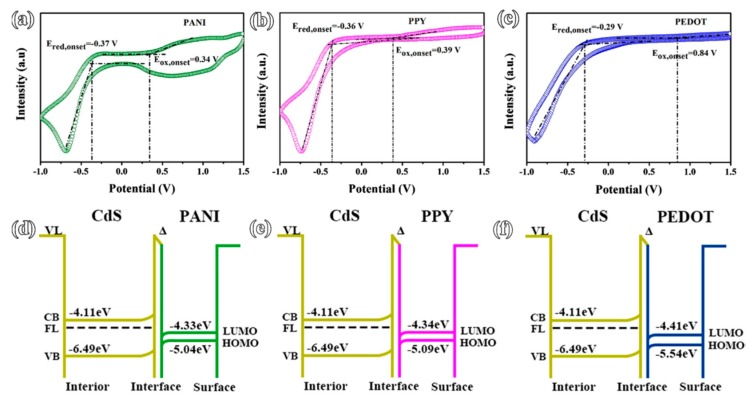
The CV curves (**a**–**c**) and band structures (**d**–**f**) of PANI@CdS, PPY@CdS, and PEDOT@CdS core-shell nanorods [49].

**Figure 8 polymers-11-00206-f008:**
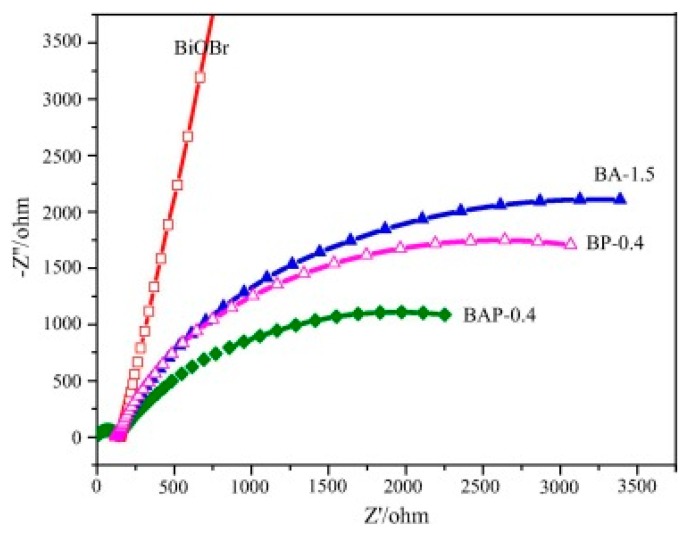
EIS Nyquist plots of BiOBr, BiOBr–Ag–1.5, BiOBr–PPy–0.4, and BiOBr–Ag–PPy–0.4 [61].

**Figure 9 polymers-11-00206-f009:**
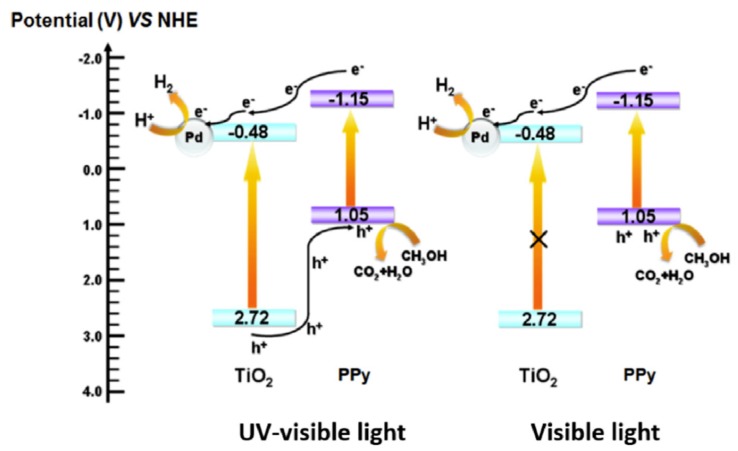
The energy levels diagram of TiO_2_–Pd–PPy photocatalysts and the photoexcited carrier transfer processes [67].

**Table 1 polymers-11-00206-t001:** Compositions, preparation methods, testing conditions, and photodegradation efficiency of PANI/bismuth-based compounds photocatalysts.

Photocatalyst	Structure	Synthesis Method	Light Source	Model Pollutants	Degradation Efficiency/Time	Ref. (Year)
PANI/Bi_12_O_17_C_l2_	Irregular nanosheets	In-situ synthesis	300-W Xe lamp (vis) (λ > 420nm)	Ciprofloxacin, Rhodamine B	CIP: 90.4%/120 min RhB: 99%/100 min	[17] (2019)
BiPO_4_-PANI	heterostructure	Hydrothermal and hybridization	300-W Xe lamp (vis) (λ > 420nm)	Methylene blue	87.3%/120 min	[18] (2018)
PANI/BiOI	Nanoplates/nanoparticles	Co-precipitation	300-W Xe lamp (vis) (λ = 420nm)	Rhodamine B	91%/120 min	[19] (2018)
PANI/BiOBr-0.2	Nanofibers	Hydrothermal	350-W Xe lamp (simulated sunlight)	Rhodamine B	99.9%/75 min	[20] (2017)
PANI/BiOCl	Nanosheets	Chemisorption	500-W high pressure Xe lamp (vis) (λ > 420nm)	Methyl orange	67%/20 min	[21] (2013)
PANI/BiVO_4_	Spindle like structure	Sonochemical	500-W Xe lamp (vis) (λ > 420nm)	Rhodamine B, phenol	RhB: 100%/60 min Phenol: 24%/120 min	[22] (2009)

**Table 2 polymers-11-00206-t002:** Compositions, preparation methods, testing conditions, and photodegradation efficiency of PANI/TiO_2_ composite photocatalysts.

Photocatalyst	Structure	Synthesis Method	Light Source	Model Pollutants	Degradation Efficiency/Time	Ref. (Year)
TiO_2/_PANI	Nanocomposites	Polymerization	Hg-lamp (UV)	Pharmaceutical: Propranolol, amitriptyline; Pesticides: sulcotrione, clomazone	Propranolol: 23%/60 min Amitriptyline: 45%/60 min Sulcotrione: 94%/60 min Clomazone: 32 %/60 min	[23] (2018)
PANI/TiO_2_-rGH	Composite hydrogel	Chemical reduction	500-W Hg-lamp (UV)	Bisphenol A	80.5%/70 min	[24] (2018)
PANI/grey-TiO_2_	Nanocomposite	In-situ polymerization	300-W Xe lamp (vis) (λ = 420 nm)	Rhodamine B	~100%/180 min	[25] (2017)
PANI/TiO_2_	Nanocomposites	In-situ polymerization	Pen-Ray UYP lamp (UVA) (315 nm < λ < 400 nm)	Reactive Red 45	94%/60 min	[26] (2017)
TiO_2_/PANI	core-shell composites	Dip-coating	45-W indoor fluorescent lamp (UV)	Reactive Red 45	85%/60 min	[27] (2014)
PANI/MS-TiO_2_	Mesoporous crystal microsphere	Solution evaporation and chemisorption	300-W Xe lamp (vis) (λ > 420nm)	Rhodamine B and Methylene blue	RhB: 99.8%/120 min MB: 99.5%/150 min	[28] (2016)
s-PANI@m-TiO_2_	Nanocomposites	In-situ polymerization	400-W lamp (vis)	Methylene blue, Brilliant blue	MB: ~100%/420 min BB: ~100%/300 min	[29] (2015)
PANI-TiO_2_	Nanotube-nanoparticles	Polymerization	(vis) (λ > 400nm)	Methylene blue	85%/300 min	[30] (2014)
TiO_2_/PANI	Microscale hierarchical 3D flowerlike	In-situ polymerization	Natural sunlight irradiation	Congo red and Methyl orange	CR: 96%/120 min MO: 90%/120 min	[31] (2014)
PANI/TiO_2_	Nanocomposites	Simple solution method	110-W high pressure Na lamp (vis) (λ > 400nm)	Methylene blue	81.74%/120 min	[32] (2010)
TiO_2_/PANI	Nanocrystalline composites	In-situ polymerization	Natural light	Methylene blue	~90%/90 min	[33] (2007)

**Table 3 polymers-11-00206-t003:** Compositions, preparation methods, testing conditions, and photodegradation efficiency of other polyaniline based composite photocatalysts.

Photocatalyst	Structure	Synthesis Method	Light Source	Model Pollutants	Degradation Efficiency/Time	Ref. (Year)
ZrO_2_/PANI	Hybrid nanocomposite	polymerization	300-W white light lamp (λ = 280 to 315 nm)	Trichlorophenol	75%/210 min	[39] (2018)
PANI/CNNS	3D hierarchical composite hydrogels	In-situ polymerization	500-W Xe lamp (vis) (λ > 420nm)	Methylene blue	89.1%/240 min	[40] (2016)
MnFe_2_O_4_/PANI	Nanocomposite	In-situ polymerization	175-W halide lamp (λ < 400 nm)	Rhodamine B	~100%/60 min	[41] (2016)
CoFe_2_O_4_/PANI	Hollow nanofibers/nanograins	In-situ polymerization	10-W LED lamp (vis)	Methyl orange	85%/120 min	[42] (2016)
ZnO/rGO/PANI	Nanocomposite	In-situ polymerization	500-W high pressure Hg lamp (UV)	Methyl orange	~100%/60 min	[43] (2016)
PANI/Ag/Ag_3_PO_4_	Composites	In-situ deposition	300-W Xe lamp (vis) (λ < 420 nm)	Rhodamine B	>95%/5 min	[44] (2014)
*g*-C_3_N_4_/PANI	Nanosheets/nanorods	polymerization	500-W Xe lamp (vis) (400 nm < λ < 700 nm)	Methylene blue and methyl orange	MB: 78.6%/30 min MO: 99.8%/30 min	[45] (2014)
Hybridized defective ZnO/PANI	Hybridized nanocrystal	Chemisorption and cold plasma treatment (CPT)	4-W UV lamp (UV) (λ = 365 nm)	Methyl orange and 4-chlorophenol	MO: ~93.6%/120 min 4-CP: ~87.8%/120 min	[46] (2014)
PANI/ZnO	Nanocomposite	Chemical oxidative polymerization	Natural sunlight (10:00 a.m. to 3.00 p.m.)	Methylene blue and malachite green	MB: 97%/300 min MG: 99%/300 min	[47] (2012)
PANI/CdS	Monolayer-hybrid	Chemisorption	500-W Xe lamp (vis) (λ > 450 nm)	Methylene blue	92%/300 min	[48] (2010)

**Table 4 polymers-11-00206-t004:** Photocatalytic hydrogen activity of PANI based composite photocatalysts.

Photocatalyst	Structure	Synthetic Method	Light Source	Sacrificial Agent	Activity (μmol h^−1^g^−1^)	Ref. (Year)
PANI@CdS	Core-shell nanorods	In-situ polymerization	PLS-SXE-300C lamp (vis) (λ ≥ 420 nm)	0.25 M Na_2_SO_3_ + 0.35 M Na_2_S	~9700	[49] (2018)
PANI/g-TiO_2_	Nanocomposites	In-situ polymerization	300-W Xe (vis)	Methanol	1790	[25] (2017)
PANI/NaTaO_3_	Nanocomposites	Polymerization	150-W Hg lamp (UV)	Formic acid	163 μmole h^−1^	[52] (2017)
PANI-ZnS	Nanocomposites	Solvothermal	Hg lamp (UV)	0.1 M Na_2_S_2_ + 0.04 M Na_2_SO_3_ + 3 M NaCl	6750 μmol h^−1^ g^−1^	[53] (2016)
Ni-ZnO/PANI	Nanorods-like	Direct impregnation	250-W Halogen visible lamp (vis)	0.2 M Na_2_S_2_O_3_ + Na_2_CO_3_	~558 558 μmole h^−1^	[54] (2014)
g-C_3_N_4_-Ag/PANI	Heterostructured nanosheets	Sonication + hydrothermal	300-W Xe lamp (vis) (λ < 420 nm)	Methanol	210.73 μmol h^−1^ mg^−1^	[55] (2017)
MoS_2_-PANI-CdS	Hybrid structure	Sonication	Day-light fluorescent lamps	0.8 M Na_2_SO_3_ + 0.6 M Na_2_S	~78 μmol	[56] (2015)

**Table 5 polymers-11-00206-t005:** Compositions, preparation methods, testing conditions, and photodegradation efficiency of polypyrrole based composite photocatalysts.

Photocatalyst	Synthesis Method	Light Source	Model Pollutants	Degradation Efficiency/Time	Ref. (Year)
PPy	Radiolysis/polymerization	300-W Xe lamp (vis) (λ < 420 nm)	Phenol, methyl orange	Phenol: 20%/300 min MO: 80%/300 min	[57] (2019)
PPy-CdS	In-situ polymerization	Xe lamp	Rhodamine B, Methylene blue	RhB: 99.34%/150 min MB: 99.93%/150 min	[58] (2018)
PPy-TiO_2_	polymerization	500-W Xe lamp (vis)	Methyl orange	100%/60 min	[59] (2017)
PPy/TiO_2_	polymerization	500-W tungsten-halogen lamp	Rhodamine B	97%/480 min	[60] (2016)
BiOBr-Ag-PPy	polymerization	150-W halogen lamp (vis) (λ < 420 nm)	Malachite green	97%/120 min	[61] (2018)
PPy-TiO_2_/M-Fe_2_O_3_	In-situ polymerization	Solar radiation	Methyl orange	59.3%/90 min	[62] (2014)
PPy-BiOI	In-situ precipitation	(vis) (λ > 420 nm)	Rhodamine B	83%/300 min	[63] (2017)
PPy/Bi_2_WO_6_	Photocatalytic oxidative polymerization	500-W Xe lamp (vis)	Phenol	~100%/120 min	[64] (2014)
PPy-ZnIn_2_S_4_	Hydrothermal	100-W Iodine-gallium lamp	Chloramphenicol	100%/60 min	[65] (2017)
PPy-AgPMo_12_	In-situ polymerization	300-W Xe lamp (vis)	Rhodamine B	73.09%/360 min	[66] (2017)

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
