# Peer review of "Recent Developments about Conductive Polymer Based Composite Photocatalysts"

_polymers, 2019, doi:10.3390/polym11020206_

Round 1
Reviewer 1 Report
Please see comments in pdf file.
PEDOT is mentioned in text only 2 times?
Table 2.
Polymerization, oxidative polymerization, in-situ polymerization and in-situ chemical oxidation are four different names for PANI-TiO2 preparation? For clarity, authors should use pick only one name. If not, please explain.

Author Response
1. Please explain
(1) p.3, green underline, “synergistic effect between the hydroxyl and superoxide radicals” (Reference 19)
Response: It has been changed to “synergistic effect between between BiOI and PANI ”.
(2) p.9, green underline, “PANI hybridized defective ZnO “ (Reference 47)
Response: It has been changed to “PANI/ZnO nanocomposite“
(3) p.14, (Reference 64)
“Based on the analysis of Raman spectra, a high resolution spectrum in the region of 100-180 cm-1 showed a red shift in wavenumber which is experienced by the Raman mode of Bi2WO6 in the composite. This analysis suggests that a covalent-like interaction between PPy and Bi2WO6 is present.”
Response: It has been changed to” Comparing the high resolution Raman spectra of Bi2WO6 /PPy and Py in the region of 100-180 cm-1, a red shift in wavenumber which is observed. The red shift suggests that there is a covalent-like interaction between PPy and Bi2WO6.”.
2. Table 2, Ref.31, ” TiO2 is prepared by sol-gel method, but TiO2/PANI composite is prepared by the chemical oxidation of aniline in presence of TiO2.”
Response: It has been changed to “In-situ polymerization”.
3. Table 1 is used only once in text. So it would be nice to see if there is some correlation between synthesis procedure and structure of the composites? Also, is there chemical interaction between PANI and model pollutants? Please see: https://pubs.acs.org/doi/abs/10.1021/jp803903x
Response: Table 1 summarize the content discussed in 2.1.1.
Some articles discuss about the interaction between PANI and semiconductor nanomaterials.
No author mentions the chemical interaction between PANI and model pollutants.
4. English
Response: Thanks for the reviewer’s suggestion about English writing.
English has been modified according to the reviewer’s comments.
5. Table 2.
Polymerization, oxidative polymerization, in-situ polymerization and in-situ chemical oxidation are four different names for PANI-TiO2 preparation? For clarity, authors should use pick only one name. If not, please explain.
Response: We pick polymerization and in-situ polymerization. In-situ polymerization means polymerization in presence of semiconductor nanomaterial.
Reviewer 2 Report
It is a very interesting and well written review article on a subject which was not thoroughly reviewed in the literature so far. Therefore, this work will be very useful to the scientific community. There are only a few general comments that have to be addressed by the authors:
· The conductive polymers referred in this article are polyaniline (PANI) and polypyrrole (PPy) only. The authors should expand their study to other conductive polymers as well, such as polythiophenes and their derivatives.
· The authors provide a lot of examples on composites based on PANI and PPy. However, there is no report on the effect of the molecular characteristics of these polymers (molecular weight and molecular weight distribution) on their performance as composite photocatalysts. In addition, in several cases there is no report on the amount of polymer in the composite and how this amount affects the performance.
Author Response
It is a very interesting and well written review article on a subject which was not thoroughly reviewed in the literature so far. Therefore, this work will be very useful to the scientific community. There are only a few general comments that have to be addressed by the authors:
1. The conductive polymers referred in this article are polyaniline (PANI) and polypyrrole (PPy) only. The authors should expand their study to other conductive polymers as well, such as polythiophenes and their derivatives.
Response: Thanks for the reviewer’s comment. A large portion of the research about conductive polymer based composite photocatalysts are polyaniline (PANI) based composites. Only limited amounts of polypyrrole (PPy), poly(3,4-ethylenedioxythiophene) (PEDOT)), and polythiophene (PTh) based composite photocatalysts were reported. Polythiophene based photocatalysts related researches were added in section 4 of the revised manuscript.
2. The authors provide a lot of examples on composites based on PANI and PPy. However, there is no report on the effect of the molecular characteristics of these polymers (molecular weight and molecular weight distribution) on their performance as composite photocatalysts. In addition, in several cases there is no report on the amount of polymer in the composite and how this amount affects the performance.
Response:
(a) According to our survey, molecular weight and molecular weight distribution were almost not mentioned in the literature. In ref 22, the molecular weight of PANI is about 100000. It was added in the revised manuscript. In addition, only one kind of polymer is used. The authors did not report on the effect of the molecular characteristics (molecular weight and molecular weight distribution) of these polymers on their performance as composite photocatalysts.
(b) We have added some details on the amount of polymer in the composite and how this amount affects the performance in several cases.
Reviewer 3 Report
I am generally very soft in my reviews but unfortunately I cannot accept this paper
1- English is unfortunately very poor
2- The paper have been selectively chosen and many important works left out
3- There is no introduction and prelude to each sub section
4- The works of references have not been analysed and they are mostly copied and pasted from their abstract
This is a poorly written review and should not be published
Author Response
I am generally very soft in my reviews but unfortunately I cannot accept this paper
1- English is unfortunately very poor
Response: English has been polished.
2- The paper have been selectively chosen and many important works left out
Response: We have done our best to search the literature. Polythiophene based photocatalysts related researches were added in section 4 of the revised manuscript.
3- There is no introduction and prelude to each sub section
Response: Introduction and prelude were added.
4- The works of references have not been analysed and they are mostly copied and pasted from their abstract
Response: We have spent lots of efforts to organize the type and structure of materials used, synthesis method, light source, model pollutants, degradation performance, H2 production performance, and mechanism for the conductive polymer based composite photocatalysts. We try to provide summarized conditions and results which can help the readers in the photocatalysis field. Some interesting finding or scheme such as the cell structure of the bacteria treated with PANI–TiO2 photocatalyst during the antibacterial test, CV curves and band structures of PANI@CdS, PPY@CdS and PEDOT@CdS core-shell nanorods, fabrication process, and mechanism were discussed. We have also added additional discussion and some other literatures according to other reviewers’ comments.
We believe the revised review article can help the researchers in the photocatalysis field.
Round 2
Reviewer 3 Report
OK for publication